Causal events enter awareness faster than non-causal events

Moors Pieter pieter.moors@ppw.kuleuven.be 1
Wagemans Johan 1
de-Wit Lee 2 3
1 Department of Brain and Cognition, Katholieke Universiteit Leuven , Leuven , Belgium
2 Institute of Continuing Education, University of Cambridge , Cambridge , United Kingdom
3 Cognition and Language Sciences, University College London, University of London , London , United Kingdom
Zohar Ada
Electronic publication date: 2017 Jan 26
Publication date: 2017
Volume: 5
Electronic Location ID: e2932
Received 2016 Nov 19; Accepted 2016 Dec 21
Copyright: ©2017 Moors et al.
Copyright year: 2017
Copyright holder: Moors et al.
License: This is an open access article distributed under the terms of the Creative Commons Attribution License, which permits unrestricted use, distribution, reproduction and adaptation in any medium and for any purpose provided that it is properly attributed. For attribution, the original author(s), title, publication source (PeerJ) and either DOI or URL of the article must be cited.
License URL: https://creativecommons.org/licenses/by/4.0/

Keywords: Continuous flash suppression, Visual awareness, Binocular rivalry, Perception of causality, Consciousness

Funding: Research Fund - Flanders (FWO) Methusalem METH/08/02 METH/14/02 This research was supported by the Fund for Research Fund - Flanders (FWO) through a doctoral fellowship awarded to PM, a postdoctoral fellowship awarded to LD, and a Methusalem grant awarded to JW (METH/08/02 and METH/14/02). The funders had no role in study design, data collection and analysis, decision to publish, or preparation of the manuscript.

==============================
Philosophers have long argued that causality cannot be directly observed but requires a conscious inference (Hume, 1967). Albert Michotte however developed numerous visual phenomena in which people seemed to perceive causality akin to primary visual properties like colour or motion (Michotte, 1946). Michotte claimed that the perception of causality did not require a conscious, deliberate inference but, working over 70 years ago, he did not have access to the experimental methods to test this claim. Here we employ Continuous Flash Suppression (CFS)—an interocular suppression technique to render stimuli invisible (Tsuchiya & Koch, 2005)—to test whether causal events enter awareness faster than non-causal events. We presented observers with ‘causal’ and ‘non-causal’ events, and found consistent evidence that participants become aware of causal events more rapidly than non-causal events. Our results suggest that, whilst causality must be inferred from sensory evidence, this inference might be computed at low levels of perceptual processing, and does not depend on a deliberative conscious evaluation of the stimulus. This work therefore supports Michotte’s contention that, like colour or motion, causality is an immediate property of our perception of the world.

Introduction

One of the first lessons we instill in statistics courses is that a ‘correlation doesn’t equal causation.’ Philosophically speaking however, the lesson should go much deeper than that. David Hume famously argued that causality was not something that could be observed at all. Even if we manipulate one variable such that it consistently leads to a certain outcome, we can use reason to infer causality from this contingency, but our senses cannot actually observe the causal interaction (Hume, 1967). Kant goes even further, and argues that causality is not a feature of the world, but is a phenomenon we experience because our minds bring the concept of causality to the world (Kant, 1783). Kant’s ideas have received renewed expression in the ‘User Interface Theory’ which claims that we experience a constructed virtual reality, which has evolved to be useful, rather than veridical (Hoffman, Singh & Prakash, 2015). The ‘virtual’ nature of our experience of the world is illustrated by the perception of colour, in which it is clear that ‘the rays, to speak properly, are not colored’ (Newton, 1704), and that colour is not inherent to a particular wavelength. Similarly causality is not inherent to particular contingencies of spatiotemporal dynamics, rather causality is an inference the mind brings to those dynamics.

Framed in this way, the key question then becomes: at what stage of mental processing is causality inferred? Albert Michotte famously argued that causality was a perceptual phenomenon, and did not require a deliberate, conscious inference (Michotte, 1946). In the 1940s, Michotte developed a series of ingenious displays which demonstrated that parametric variations in the spatiotemporal attributes of a stimulus could predictably influence whether a causal percept was elicited. In one classic example, Michotte presented observers with a launching event in which an object moves toward another stationary object, and stops right in front of it. The second object then immediately starts to move along the same trajectory (Fig. 1A). This event almost irresistibly evokes a causal impression, in which the first object appears to cause the motion of the second object. Based on this observation, and a series of meticulous experiments with critical parametric variations, Michotte argued that human observers perceive causality, and that causality is akin to a primary visual property such as colour or motion. In the seventy years following his seminal contribution, the discussion on the perceptual versus cognitive basis of causality perception has remained wide open (Weir, 1978; Scholl & Tremoulet, 2000; Wagemans, Van Lier & Scholl, 2006; Wolff, 2007; Rips, 2011; Rolfs, Dambacher & Cavanagh, 2013; Arnold et al., 2015). Whilst the demonstrations developed by Michotte do appear compelling, it is hard (if not impossible) to know whether one’s conscious thoughts and intentions—whilst watching the stimuli—shape or influence the way in which we perceive them, particularly given that participants are fully aware of the display, and are primed to think about, and indeed make judgements about causality (Choi & Scholl, 2006).

Figure 1 (A) Events used in both experiments. In the launch event (Experiments 1 and 2), a disc starts moving toward a stationary disc, stops right in front of it, and the stationary disc then starts moving. In the pass event (Experiments 1 and 2), a disc starts moving toward a stationary disc, stops when it fully overlaps with the stationary disc, and the stationary disc then starts moving. In the pseudo-launch event (Experiment 2), a disc starts moving toward a stationary disc, and stops after passing by the side of the stationary disc, after which the stationary disc starts moving. (B) Trial sequence used in the experiments. After a fixation period (1 s), the moving CFS mask was presented to the dominant eye and the (non-) causal event to the other eye. The disc events repeated and gradually increased in contrast until observers detected any part of the discs. In the second frame, the first disc is not drawn as a full disc because it appears from behind the virtual occluder.

Here we seek to further Michotte’s work by exploiting modern experimental methods to ensure participants are not initially conscious of the stimuli presented to them, and importantly never have to make causality judgments on the stimuli. More specifically, in two experiments, we used a modified version of Continuous Flash Suppression (CFS) (Tsuchiya & Koch, 2005; Moors, Wagemans & De-Wit, 2014) that is able to render moving events invisible for extended periods of time, by presenting a dynamic noise pattern to one eye (Fig. 1B), suppressing the stimulus presented to the other eye. The particular implementation of CFS employed here is a paradigm also known as the breaking CFS or b-CFS paradigm (Stein, Hebart & Sterzer, 2011). Here, initially suppressed stimuli are gradually increased in contrast during a certain initial time window and then presented at maximal contrast until they “break into” awareness as measured by participants’ detection time with respect to a certain stimulus attribute (e.g., its location (left/right or up/down) relative to the fixation cross). We reasoned that if a differential sensitivity to causal and non-causal events is revealed through CFS, this would imply that the visual system differentially processes these events in the absence of a conscious inference. It is important to stress that the “absence of a conscious inference” does not pertain to so-called “unconscious processing” of the causal nature of these events. Rather, it refers to the fact that participants are not required to consciously evaluate the causal nature of the events they are presented with. Although the paradigm employed in this study could be extended to address this question about unconscious processing of causal events, the use of suppression times in principle allows us only to measure any kind of differential sensitivity to the causal nature of the events, be it due to pre-conscious or conscious processing (Stein, Hebart & Sterzer, 2011; Stein & Sterzer, 2014). Note that the use of the so-called “binocular control condition” (Jiang, Costello & He, 2007) does not ameliorate this problem. Therefore, we did not implement such a condition in both experiments reported here.

Experiment 1

In our first experiment, we presented observers with launching (causal) and passing (non-causal) events while these were rendered invisible through CFS. The launching event contained two discs, the first disc would start moving towards a second stationary disc, stop right in front of it, at which point the second disc would start moving. The passing event also involved two discs, but the first disc completely overlapped the second disc before the latter started moving (Fig. 1A) (Scholl & Nakayama, 2002). At trial onset, these events were invisible to participants and they were continuously repeated until participants detected any aspect of the discs emerging through the CFS mask (i.e., observers responded as soon as they saw one or more discs). Importantly, the task was completely unrelated to the causal nature of the events.

Methods

Participants

We recruited 24 participants, of which 18 were included in the final analysis. The other six participants were excluded because of exclusion criteria defined further below. All had normal or corrected-to-normal vision and participated in return for course credit or a monetary compensation. The study was approved by the local ethics committee of the faculty (the Social and Societal Ethics Committee of the KU Leuven (SMEC) under the approval number G-2014 08 033). All participants provided written informed consent prior to the start of the experiment. As we did not have a good intuition about the size of any potential effect, we did not specifically determine our sample size before the start of the experiment. Rather, we decided to collect (what is considered to be) a decent number of participants (∼25).

Apparatus

Stimuli were shown on two 19.8-in. Sony Trinitron GDM F500-R (2,048 × 1,536 pixels at 60 Hz, for each) monitors driven by a DELL Precision T3400 computer with an Intel Core Quad CPU Q9300 2.5 GHz processor running on Windows XP. Binocular presentation was achieved by a custom made stereo set-up. Two CRT monitors, which stood opposite to each other (distance of 220 cm), projected to the left and right eye respectively via two mirrors placed at a distance of 110 cm from the screen. A head- and chin rest (15 cm from the mirrors) was used to stabilize fixation. The effective viewing distance was 125 cm. Stimulus presentation, timing and keyboard responses were controlled with custom software programmed in Python using the PsychoPy library (Peirce, 2007; Peirce, 2009).

Stimuli

The background of the display in both eyes consisted of a random checkerboard pattern to achieve stable binocular fusion. The individual elements of the checkerboard were 0.34° by 0.34°. In both eyes, a grey frame (10° by 10°) where the stimuli were presented was superimposed on the checkerboard pattern. A black (eye dominance measurement) or white (main experiment) fixation cross was presented at the start of each trials for 1 s (size 0.5° by 0.5°). In the eye dominance measurement phase, the target consisted of an arrow (maximal width 4°, maximal height 2°) and the CFS mask consisted of 150 squares with randomly selected sizes between 1° and 2° and a random luminance value. Two grey discs (1° of visual angle) were used to generate the launch and pass events. The starting position of the first disc was located at one of 8 different positions along a virtual circle (2.5° and 3.5° radius for the pass and launch events, respectively) at equally spaced angles (0–270°, in steps of 45°). This manipulation of event direction was used to ensure that participants would not adapt to specific event directions over the course of the experiment, and thus to provide sufficient variation with respect to the stimuli that were presented. The starting position of the second disc was always in the centre of the virtual circle. The centre of the virtual circle was jittered on each trial (in a range of ±0.5° horizontally and vertically). The discs moved at a speed of 5°/s. As the duration for the launch event would be shorter compared to the pass event for a fixed trajectory length, we adjusted for this by increasing the total distance travelled for the launch event by 1°. We adjusted the length of the trajectory, rather than the speed at which the discs moved as this could induce saliency differences between the events to which we know CFS is sensitive (Moors, Wagemans & De-Wit, 2014). The first disc always appeared from behind a virtual occluder and the second disc would disappear behind a virtual occluder as the event sequence ended after 1,000 ms. To ensure a continuous flow of events, 100 ms after the first event sequence, the event initiated again in the reverse order. That is, what previously was the second disc now would appear from behind a virtual occluder, approach what previously was the first disc, and this disc would disappear again behind a virtual occluder. This event loop continued until the participant’s response. In the case of a launch, the first disc always stopped just before the second disc after which the second disc started moving, yielding a percept of the first disc causing the movement of the second disc. For a pass event, the first disc stopped when it fully overlapped with the second disc after which the second disc started moving, yielding a percept of the first disc passing over the second disc and completing its trajectory.

The CFS mask (8° × 8°) consisted of 104 moving squares (Moors, Wagemans & De-Wit, 2014) with a random luminance value and size (uniform range 0.2–2°), moving at a speed between 3°/s and 7°/s (randomly determined for each element using a uniform range). The mask elements would be partially occluded if they moved outside of the 10° × 10° frame. If they disappeared completely from behind one side of the frame, they would be moved behind the opposite side of the frame continuing on the same trajectory.

Procedure

In the first part of the experiment, participants’ eye dominance was measured according to the procedure outlined by Yang, Blake & McDonald (2010). Here, on each trial, a CFS mask was presented to one eye while an arrow pointing either to the left or the right was presented to the other eye and gradually increased in contrast. Upon breakthrough of the arrow stimulus, the participant had to indicate its direction as quickly as possible. Eye dominance was then determined by comparing the mean suppression times for both eyes, and the eye for which the CFS mask elicited the longest mean suppression times was taken as the dominant eye. In all subsequent parts of the experiment, the CFS mask was presented to this eye.

In the main part of the experiment, participants were presented on each trial with a moving CFS mask in one eye and a launch or pass event in the other eye (presented in one of eight directions). The discs started at 3% contrast at the beginning of the trial and the contrast was increased by 3% after each iteration of the event sequence. The participants’ task was to indicate as quickly as possible when they perceived one or more discs that became visible among the moving squares. We included catch trials in which no event was presented. These catch trials self-terminated after 10 s, upon which a new trial was initiated. Experimental trials were always presented until breakthrough. Participants were told before the start of the experiment that trials could sometimes self-terminate, and that this was a characteristic of the experiment to ensure trials would not take too long to complete. Before starting the main experiment, participants completed a practice block to become acquainted with the task.

Design

The experiment consisted of a 2 × 8 within-subjects design with the factors event type (launch vs. pass) and event direction (eight different, evenly spaced, directions). Participants completed 192 experimental trials in total. 48 catch trials were included. All conditions were randomized across trials. The practice block consisted of 16 trials.

Data-analysis

All analyses were conducted in R 3.2.0, a statistical programming language (R Core Team, 2014). All statistical tests were performed in the Bayesian framework relying on model selection through Bayes Factors, using the R BayesFactor package (version 0.9.12-2) (Morey & Rouder, 2015). In short, the Bayes Factor quantifies how consistent the data are with the predictions of one statistical model relative to those of another and therefore provides an intuitive measure to quantify the degree of belief in one statistical model over another (e.g., a model with and without a main effect of event type). All fitted models were ANOVA-style models including random intercepts for participants, and the default settings for the priors (medium prior scale for fixed effects and nuisance prior scale for the participant random effect). We used the variables “event type” (launch vs. pass) and “event direction” (the 8 directions were recoded into horizontal vs. vertical vs. oblique) as predictors in the analysis. As a guideline to interpret the resulting Bayes Factors, we use the classification proposed by Jeffreys (1961) in that Bayes Factors from 3 onwards constitute substantial evidence for one model over the other. The data for Experiments 1 and 2 are available at https://figshare.com/s/cb4521afc14ced461fdc.

Results and discussion

All participants that responded on more than five (∼10%) catch trials were removed from the data (n = 2). Furthermore, because we wanted to analyse suppression times that were recorded after the first appearance of an event (which was after 500 ms), we excluded all trials in which the suppression time was shorter than 800 ms (i.e., we took a minimal response time of 300 ms and added that to the time point at which the discs interacted to determine the threshold). This ranged from deleting no trials to 25% of trials (M = 4.2, SD = 7.3). If more than 10% of the participants’ data needed to be removed due to this, we removed the participant from the analysis (n = 4). Additionally, we defined outliers as suppression times that deviated more than three standard deviations from the participants’ mean suppression time and removed these observations. This ranged from deleting no trials to 2% of trials (M = 1.1, SD= 0.55). Please note that including outliers, or including the participants that did not perform well on the catch trials or responded too fast on more than 10% of trials, does not change the outcome of the analyses. We refer to the Supplemental Information 1 for an analysis in which we assess the robustness of the results based on all combinations of exclusion criteria (exclusion based on percentage responses on catch trials, exclusion based on percentage too fast trials, exclusion of outliers). Due to their positive skewness, suppression times were logarithmically transformed at the participant level, before subjecting them to any analysis.

The results indicated that launch events (M = 3.86 s, SD = 1.50) entered awareness faster than pass events (M = 4.14 s, SD= 1.57) (BF > 100 for a model including an effect of event versus an empty model containing only random participant variability), for nearly every observer (Fig. 2). Table 1 summarizes the results of the Bayes Factor analysis. As can be derived from the Table, a main effect of event direction was also observed. This main effect was mostly due horizontal event directions (M =3.70 s, SD = 1.37) entering awareness faster compared to the vertical (M = 4.09 s, SD = 1.69) and oblique (M = 4.11 s, SD = 1.56) directions.

Figure 2 Results of Experiment 1.

(A) Bar plot depicting mean suppression times for both event types. Errors bars are 95% within-subject confidence intervals. (B) Plot depicting the effect of event type for each observer separately (gray) and across all observers (black, difference between the bars in the left figure). Dots depict the mean difference between suppression times from pass and launch events, where positive values indicate that launch events entered awareness faster compared to pass events. Lines indicate 95% bootstrapped confidence intervals (based on 10,000 bootstrap samples). Almost all dots lie to the right of a zero difference, indicating that launch events enter awareness faster than pass events for nearly every observer.

Table 1 Bayes Factor analysis of Experiment 1.

Model	Bayes Factor	Error	
Condition + Direction	1	0	
Condition * Direction	37	0.03	
All other models	>100	NA	
Notes.

All Bayes Factor were computed with the best fitting model (top row) in the numerator and the model under consideration in the denominator. All reported Bayes Factors thus can be interpreted as how much more consistent the data are with the predictions made by the best fitting model compared to the model under consideration.

These results indicate that the visual system is sensitive to the causal structure of events in such a way that this sensitivity influences the pace at which these events enter awareness, when they are initially rendered invisible. Whilst this difference could reside in the potential to interpret one event as causal, the launch and pass events differ in a number of other ways, amongst which an aspect we refer to as the local motion contrast. That is, although pass events were drawn such that the first disc stopped when it completely overlapped with the second disc prompting the motion of the second disc, perceptually these events consist of a continuous motion signal. In contrast, launch events contain sharp motion on- and offsets. Indeed, it is known that low-level differences in contrast, spatial frequency, or orientation sometimes drive observed differences between suppression times (Gray et al., 2013; Moors et al., 2016b). In our second experiment, we sought to replicate the difference found in the first experiment in an independent sample, and included an additional control condition for the difference in the local motion contrast between the launch and pass event. This additional control condition (referred to as a “pseudo-launch” event) was exactly the same as the launch event (in terms of stimulus energy), except for the starting position of the first disc, which was shifted such that it would stop after passing by the side of the second stationary disc. This means that the secondary disc would only start moving after the first disc had stopped in a location where it could not logically have caused the motion of the second (Fig. 1A), while the event contains the same number of sharp motion on- and offsets.

Experiment 2

Methods

Participants

We recruited 27 new participants, of which 17 were included in the final analysis. One participant failed to complete the full experiment, and nine others were excluded because of the exclusion criteria used during data analysis. All participants had normal or corrected-to-normal vision and participated in return for course credit or a monetary compensation. The study was approved by the local ethics committee of the faculty (the Social and Societal Ethics Committee of the KU Leuven (SMEC) under the approval number G-2014 08 033). All participants provided written informed consent prior to the start of the experiment. Because the effect observed in Experiment 1 appeared quite strong (according to the Bayes factor analysis), we decided to collect a similar number of participants in Experiment 2.

Apparatus

The experimental set-up was exactly the same as in Experiment 1.

Stimuli

The stimuli were exactly the same as in Experiment 1, except for the pseudo-launch event described here. The pseudo-launch event was modified from the sequence of the launch event. A pseudo-launch event was generated by shifting the starting position of the first disc upward or downward (relative to the second disc, orthogonal to the direction of motion of the disc) by the size of the disc. The starting position of the first disc was also shifted forwards two times the size of the disc in the direction in which it would travel, such that during its movement it would pass next to the second disc, and stop just after it had passed by the side of the second disc. The second disc would then move in exactly the same manner in which it moves in the launch event.

Procedure

The procedure was exactly the same as in Experiment 1.

Design

The experiment consisted of a 3 × 8 within-subjects design with the factors event type (launch vs. pass vs. pseudo-launch) and event direction (eight different, evenly spaced, directions). Participants completed 192 experimental trials in total. A total of 48 catch trials were included. All conditions were randomized across trials. The practice block consisted of 16 trials.

Results and discussion

We used the same data processing pipeline as in Experiment 1. All participants that responded on more than five (∼10%) catch trials were removed from the data (n = 5). All trials in which the suppression time was shorter than 800 ms were also excluded. This ranged from deleting no trials to deleting 60% of trials (M = 6, SD = 12.9). Five participants exceeded the threshold of more than 10% trials shorter than 800 ms. One of these also fulfilled the criterion to be excluded based on responding to more than 10% of catch trials. Thus, the total number of participants that was removed based on the exclusion criteria was equal to 9. Additionally, we excluded outlying suppression times defined as deviating more than three standard deviations from the participants’ mean suppression time. This ranged from deleting no trials to deleting 2.6% of trials (M = 1.1, SD = 0.8). Again, including these outlying trials, or including the participants that did not perform well on the catch trials or had too many fast responses, does not change the outcome of the analyses (see Supplemental Information 1), and only strengthened the evidence for the pattern of results reported here. Logarithmically transformed suppression times were again the unit of analysis, and event direction was recoded in the same way as in Experiment 1.

The results of Experiment 2 replicate the result of the first experiment showing that launch events (M = 3.58 s, SD = 1.21) entered awareness faster than pass events (M = 3.82 s, SD = 1.34) (see Fig. 3). Critically, launch events also entered awareness faster than pseudo-launch events (M = 3.75 s, SD = 1.21). Table 2 summarizes the results of the Bayes Factor analysis, which are very much in line with the results of Experiment 1. Indeed, the best fitting model is exactly the same as in Experiment 1. The main effect of event direction was again due to horizontal event directions (M = 3.42 s, SD = 1.04) breaking suppression faster compared to vertical (M = 3.94 s, SD = 1.44) and oblique event directions (M = 3.76 s, SD = 1.27). Furthermore, paired Bayesian t-tests indicated that launch events entered awareness faster than pass events (BF >100) and pseudo-launch events (BF = 3), while the data appeared to be most consistent with the absence of a suppression time difference for pass and pseudo-launch events (BF < 0.1).

Figure 3 Results of Experiment 2.

(A) Bar plot depicting mean suppression times for all three event types. Errors bars are 95% within-subject confidence intervals. (B) Plot depicting the effect of event type for each observer separately (gray) and across all observers (black, difference between the bars in the left figure). Dots depict the mean difference between suppression times from pass and launch events, where positive values indicate that launch events entered awareness faster compared to pass events. Squares depict the same information, but now for launch vs. pseudolaunch events. Lines indicate 95% bootstrapped confidence intervals (based on 10,000 bootstrap samples). Most of the data points lie to the right of a zero difference, indicating that launch events enter awareness faster than pass or pseudolaunch events.

Table 2 Bayes Factor analysis for Experiment 2.

Model	Bayes Factor	Error	
Condition + Direction	1	0	
Direction	33	0.024	
All other models	>200	NA	
Notes.

All Bayes Factor were computed with the best fitting model (top row) in the numerator and the model under consideration in the denominator. All reported Bayes Factors thus can be interpreted as the how much more consistent the data are with the predictions made by the best fitting model compared to the model under consideration.

Thus, the results of Experiment 2 indicated that the suppression time difference between launch and pass events was confirmed in an independent sample. Furthermore, the addition of the pseudo-launch event revealed that this type of event entered awareness as fast as pass events did rather than launch events. This indicates that the local motion contrast was not exclusively determining the suppression time difference observed in Experiment 1, and puts more weight to interpreting the suppression time differences being due to the differences in the causal structures of the events.

Discussion and Conclusion

The results of Experiments 1 and 2 both suggest that the human visual system shows differential sensitivity to events that are more or less able to elicit causal percepts. This suggests that whilst our senses clearly cannot directly register the ‘…connexion betwixt causes and effects’ (Hume, 1967), the ability to compute a difference between events that are more or less likely to elicit causal percepts occurs independent of observers having to explicitly report their percept of the event. This result is very consistent with Michotte’s claim that ‘there is actual perception of causality, in the same sense that there is perception of shapes, movements (il y a veritablement perception de la causalité, au meme titre qu’il y a perception de formes, de mouvements)’ (Michotte, 1946). In other words, just as we do not have to consciously infer that something is ‘round’ or ‘fast,’ it seems we do not have to consciously reflect for the visual system to compute the difference between events that more or less likely to elicit causal percepts. This conclusion had been suggested by Michotte’s work, but these previous demonstrations and experiments were always limited by the fact that participants were aware of the stimuli, primed to think about them, and make judgements in terms of causality (Choi & Scholl, 2006), a term not even mentioned to participants during the experiments.

As such, our results provide additional evidence for the claim that the causal structure of events is encoded as a basic and elemental feature by the visual system. Indeed, along these lines, a recent study showed that it is possible to measure adaptation aftereffects after prolonged exposure to launch events. Furthermore, this adaptation aftereffect was retinotopically specific lending further support to the notion that the aftereffect is perceptual in nature and that neuronal populations might exist that are dedicated to the encoding of causal events (Rolfs, Dambacher & Cavanagh, 2013)—but see Arnold et al. (2015) for contrasting views on this finding. Second, neuroimaging studies have provided evidence that the neural signature associated with the causality of events is independent from attending these events or not or having to explicitly judge the causality of the events, suggesting that the causal structure of events is a basic feature encoded by the visual system (Blakemore et al., 2001; Fugelsang et al., 2005; Roser et al., 2005). Third, the perception of causality develops already very early in infancy (Saxe & Carey, 2006). Last, similarity judgments of launching events that vary in their spatial and temporal properties (i.e., spatial and temporal gaps between both objects) indicate that the classic launching stimulus is easiest to discriminate from other launching stimuli, highlighting its distinctive nature during visual processing (Young & Sutherland, 2009).

Our results can also be framed to be informative about what is processed during perceptual suppression induced by CFS. That is, although initial b-CFS studies seemed to provide very strong evidence for elaborate and complex unconscious processing of the suppressed stimulus (Jiang, Costello & He, 2007; Mudrik et al., 2011; Sklar et al., 2012), evidence has been accruing that processing of a stimulus rendered invisible through CFS is largely restricted to basic visual features such as orientation, spatial frequency, color, curvature, contrast energy, and spatial coherence (Gray et al., 2013; Hedger, Adams & Garner, 2015; Moors et al., 2016a; Moors et al., 2016b; Moors, Wagemans & De-Wit, 2016). Furthermore, neural processing of the invisible stimulus has been shown to be largely restricted to early visual areas (Hesselmann & Malach, 2011; Yuval-Greenberg & Heeger, 2013; Fogelson et al., 2014). Together with the neuroimaging evidence on the perception of causal events, we speculate that the observed advantage for causal events could stem from early levels of visual processing, basically extracting the features of the retinal stimulation and organizing them, before deliberately making sense of them.

On a more cautionary note, whilst we can say that the visual system appears to display a heightened sensitivity to events that are known to be more or less able to elicit causal percepts, we cannot definitively prove that this is because the visual system has made an inference that one event is causal and one not. It could be that early visual processing identifies some spatiotemporal properties as a ‘proto-causal’ representation, which requires further elaboration before a truly causal inference is made. With this limitation in mind however, we argue that this result provides evidence that further strengthens Michotte’s claim that causality, like shape or motion, is a basic feature of our perception.

Supplemental Information

Supplemental Information 1 Supplementary Analysis

Click here for additional data file.

We would like to thank San Verhavert for assistance in data collection.

Additional Information and Declarations

Competing Interests

Author Contributions

Human Ethics

Data Availability

The authors declare that they have no competing interests.

Pieter Moors conceived and designed the experiments, performed the experiments, analyzed the data, wrote the paper, prepared figures and/or tables.

Johan Wagemans and Lee de-Wit conceived and designed the experiments, wrote the paper, reviewed drafts of the paper.

The following information was supplied relating to ethical approvals (i.e., approving body and any reference numbers):

Social and Societal Ethics Committee of the KU Leuven (SMEC) approval: G-2014 08 033.

The following information was supplied regarding data availability:

https://dx.doi.org/10.6084/m9.figshare.3543503.v1.

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
