# Peer review of "Causal events enter awareness faster than non-causal events"

_PeerJ, doi:10.7717/peerj.2932_

## Round 0.1 · original submission · Major Revisions

Your manuscript has now been reviewed by three experts and although they required revisions, they are all positive about this paper.

Please address their concerns and summarize your revisions in order in the rebuttal letter so it is easy to follow.

All the best
Ada

·

Basic reporting

The article is very well written, and draws from both experimental psychology and philosophy in order to establish the research question and its meaning – which I find crucial for dealing with questions of this type. I found it clear and compelling, both with respect to writing and structure. I was also impressed by the cautionary approach taken by the authors in discussing and interpreting their findings – their final paragraph, for example, obviated my main concern about the possible interpretations of these results.

Experimental design

The research question is well defined and the methods are overall solid. I did have some questions and comments about the design – some pertain to the actual experiments, and some – to the way in which they were described.

1. I did not understand the display loop. Was the same event presented over and over again, starting from the very same location on the virtual circle? What does it mean that it was presented in the “reverse order”? Is it that now circle B started to move towards circle A that was now in the middle? How long was each such event? I only saw that the gaps between every two events was 100ms.
2. The methods imply that there were only 12 trials per experimental condition in Exp 1 and 8 in Exp 2 (type X direction). This seems very low in terms of signal to noise ratio, especially when dealing with subliminally presented stimuli. Indeed, it seems like eventually these eight directions were collapsed into 3 classes – but then wouldn’t there be a substantially different number of trials per direction (12/8 trials for vertical, 12/8 trials for horizontal, 72/48 for all other directions which are all oblique)?
3. What determined the sample sizes? Why 24 subjects in the first experiment and 27 in the second? Only after reading both results section it became clear that the authors wanted to have a similar N (22), which was the final sample size after excluding subjects. It is not very clear why 22, and in any case I would write it differently so in the methods the final sample size will already be clear (“22 subjects were included… additional /5 were excluded due to...”).
4. I think that the “pass” condition is not clear enough; if I understood correctly, due to the overlap between the circles, it looks as if the first circle is actually passing and completing its move, rather than causing the movement of the second circle. But this was not clearly stated in the text or in the legend of Figure 1. I would say it explicitly, to facilitate the reader’s understanding.
5. Figure 1 is not very clear. I would add a black framework around each frame (panel B) to make them more separable from one another. Also, in the second frame, the circle looks cut – is it because of the visual occluder? If this is the case, I would explain in the legend since otherwise it just looks strange.

Validity of the findings

The authors report two experiments in which causal-like scenarios emerge into awareness faster than non-causal scenarios. The effect is replicated twice, which is highly important, especially in this field, and I am very favorable of the Bayesian approach the authors take. Again, I did have some concerns and suggestions about the analyses.
1. In the provided datasets, some trials are listed as outliers. Yet in the manuscript this is not mentioned. How were these defined? Did the results pertain also without removing these outliers?
2. Trial exclusion – 500ms response time sounds like a very long estimation. Subjects can respond after 300ms and sometimes even less. So I would set the threshold at 800ms. In any case, if there was a subject that responded 38% prior to what can be reasonably expected – so that he/she responded irrespective of the event (before it could be processed) – that subject should be removed. It is not enough to remove the problematic trials only. Also, in Exp. 2 there are no data about the range of trials that were excluded.
3. The authors state that the effect was manifested for nearly every observer, but figure 2 suggests otherwise: 10-11 subjects seem to be right on the diagonal or very slightly above it (and an additional one is below it). I would either change the text of the figure – possibly adding a center point for each subject. An even more meaningful way to present the results would be to add SEs per subject, so the reader could estimate how far each subject is from the diagonal.

·

Basic reporting

The manuscript is well written. The motivation and background for the study are described in a clear and comprehensible way. Furthermore, the authors made their raw data publicly available. I really enjoyed the authors’ introduction, especially the authors’ approach to revive an older question that can now be studied with newly developed methods like CFS.

1) One point I found a bit imprecise in the introduction: The authors write that for breaking-CFS the stimulus contrast is increased until the stimulus breaks into awareness. While this is certainly true in the authors’ case, usually in breaking-CFS studies, as far as I am aware, the stimulus contrast is ramped up during an initial time window and then presented at full contrast until the observer indicates to perceive the stimulus.

2) Could the authors please also include the fixation cross in Figure 1B that was presented during CFS?

Experimental design

The experimental design is suitable to address the authors’ research question. I only have some minor comments regarding some details that could be added in the Methods section.

1) Could the authors please report the duration of a single event in the Methods section? As far as I can see, it is only mentioned once in the Results section.

2) Was there a maximum trial duration upon which trials were aborted without participants’ response (apart from the catch trials)?

3) What was the reason for showing the events in reversed order after the previous event was finished? I guess this was done to avoid sudden onsets of the stimuli at the beginning of each sequence. If so, or if there is another reason, could the authors please explicitly mention this in the manuscript?

4) In ll. 145f, the authors note that in pass events the first disc stopped and then the second disc started moving. This sounds to me as if there were motion on- and offsets also during pass events (and not only during launch events). In the Discussion section of Experiment 1, it is mentioned, however, that there was a continuous motion signal in pass events (ll. 235f).

5) Did any of the participants from Experiment 2 also participate in Experiment 1?

Validity of the findings

The authors used a well-controlled experimental design. The statistical analysis is sound and state-of-the-art. There is one point, however, regarding the mechanism underlying the observed effect that I would like to comment on.
In contrast to ‘typical’ breaking-CFS studies, the authors do not show a static stimulus, but a continuous sequence. However, not every part of this sequence is relevant with respect to the question of the influence of causality on perception. Only from the middle part of each event, causal events can be distinguished from non-causal events. The question is thus, whether causal events access awareness shortly after the critical time point (that is, during the middle part of an event), while non-causal events access awareness rather during the end phase of an event. Alternatively, it could be that causal events access awareness already at a lower contrast, while non-causal events enter awareness in a following sequence when the contrast has already been increased. Could the authors distinguish between these two possibilities on the basis of their data? For instance, the authors could bin their data according to the sequences (i.e. into 500 ms bins) and analyse whether responses to causal events fall into an earlier time bin in contrast to non-causal events. Maybe this could make it easier to infer the underlying mechanism. I understand, however, if such an analysis is not feasible, for instance due to the uncertainty regarding participants’ response times.

Reviewer 3 ·

Basic reporting

This paper is well-written and very clear. The background is comprehensive, and the structure works well.

Experimental design

I wondered whether b-CFS was the best measure to use for this experiment, however, the authors have justified it well. They suggest that they’ve used the b-CFS task to infer whether differences between conditions occur in the absence if conscious inference, rather than using it in the standard way – to infer which stimuli can and cannot be processed outside of conscious awareness. This is important, as suggested by the authors, because it’s difficult to know how much the b-CFS RT measure can really be attributed to unconscious processing. I do think the authors should try to be very clear on this point in the abstract, given that some will assume b-CFS has been used to address the question of unconscious processing (particularly given the title). It is, nonetheless, much clearer from the introduction and discussion.

Validity of the findings

I find the results compelling – the replication with the additional control is important, and the Bayesian analyses are well-explained. However, I’m slightly confused by the temporal dynamics of the events that are displayed, and how they correspond to the RTs in the CFS task. Not many studies have used CFS to suppress dynamic stimuli, and I’m not entirely sure what the RTs in this case actually reflect, given that the stimuli are repeated until a response is made. In all sequences, I assume only the ‘middle of the event’ section is particularly important, as the start phase and end phase are similar in all conditions, and do not contain much information on causality. It would be helpful for the authors to comment on this, including:
• How long did each event last?
• How does the length of the event correspond with the RTs? i.e. how many events were needed before the stimuli became visible? Were the RTs time-locked in any way to the salient 'middle section' of the event?

---

## Round 0.2 · accepted · Accept

Dear Authors,

In this case the reviewers from the first round were in agreement (not always the case...) and you addressed all their concerns. This is a very interesting piece of work! Cheers!